# Rapid Maxillary Expansion (RME): An Otolaryngologic Perspective

**DOI:** 10.3390/jcm11175243

**Published:** 2022-09-05

**Authors:** Luca Cerritelli, Stavros Hatzopoulos, Andrea Catalano, Chiara Bianchini, Giovanni Cammaroto, Giuseppe Meccariello, Giannicola Iannella, Claudio Vicini, Stefano Pelucchi, Piotr Henryk Skarzynski, Andrea Ciorba

**Affiliations:** 1ENT and Audiology Unit, Department of Neurosciences and Rehabilitation, University Hospital of Ferrara, 44124 Ferrara, Italy; 2Department of Head-Neck Surgery, Otolaryngology, Head-Neck and Oral Surgery Unit, Morgagni Pierantoni Hospital, Via Carlo Forlanini, 47121 Forlì, Italy; 3Institute of Sensory Organs, 1 Mokra Street, 05-830 Nadarzyn/Kajetany, Poland; 4Institute of Physiology and Pathology of Hearing, 10 Mochnackiego Street, 02-042 Warsaw, Poland; 5Department of Heart Failure and Cardiac Rehabilitation, Medical University of Warsaw, 8 Kondratowicza Street, 03-242 Warsaw, Poland

**Keywords:** children, rapid maxillary expansion, oral breathing, obstructive sleep apnea, adenotonsillectomy

## Abstract

*Background***.** To evaluate the possible effects of Rapid Maxillary Expansion (RME), such as nasal breathing problems, middle ear function, Obstructive Sleep Apnea (OSA) in the otolaryngology field. RME has already been introduced in orthodontics to expand the maxilla of young patients affected by transversal maxillary constriction. *Methods***.** A literature search was performed using different databases (Medline/PubMed, EMBASE, and CINAHL), from May 2005 to November 2021, according to the PRISMA guidelines. *Results***.** The application of RME in children has shown good results on nasal function, reducing nasal resistances, independently from a previous adenotonsillectomy. These results are not only related to the increasing of nasal transverse diameters and volume, but also to the stiffening of airway muscles, enabling the nasal filtrum function and avoiding mouth opening, thereby decreasing respiratory infections. Positive effects have also been reported for the treatment of conductive hearing loss and of OSA, with the reduction of Apnea Hypopnea Index (AHI), possibly due to (i) an increased pharyngeal dimensions, (ii) a new tongue posture, and (iii) reduced nasal respiratory problems. *Conclusions***.** Otolaryngologists should be aware of the indications and benefits of the RME treatment, considering its possible multiple beneficial effects.

## 1. Introduction

Rapid maxillary expansion (RME) has been used as a routine clinical procedure in orthodontics in children, to expand the maxilla of young patients presenting transversal maxillary constriction, deep palatal vault, and accompanying cross-bite and crowding, until the palatal suture is not fully ossified [1]. 

This treatment is performed by applying an expansion screw welded to the bands on the first premolars and first molars or with similar applications. The expansion screw is activated daily and forces the mid-palatal suture to open and the maxillary bones to diverge from each other. In particular, RME increases the transverse dimensions of maxilla and nasal cavity by separating the two maxillary halves from the mid-palatal suture in a short period.

Data from the literature suggest that potential favorable effects of RME are available on nasal breathing, recurrent ear or adenoidal infections, OSA or on the voice quality [1].

The purpose of this paper is to review the studies investigating RME effects on otolaryngologic issues, such as naso-respiratory problems, middle ear function and obstructive sleep apnea (OSA).

## 2. Materials

The PubMed, Embase and Cinahl databases were searched for the last 16 years (from November 2005 up to November 2021). Full-text articles were obtained in cases where the title, abstract, or key words suggested that the study may be eligible for this review. The medical subject heading (MeSH) terms included: Rapid maxillary expansion (RME) in children, RME and obstructive apnoea in children, RME and conductive hearing loss in children, RME and oral breathing, RME and sleep in children, RME and nasal resistance, RME and nasal width. 

The search was conducted according to the PRISMA criteria/guidelines (http://www.prisma-statement.org/ (accessed on 16 July 2022): it was carried out independently and restricted to papers in English. Initially, the total number of papers identified containing the words “RME in children” was 725, from which we excluded articles published before 2005 (*n* = 172).

Inclusion criteria were clinical series and review papers. Exclusion criteria were the following: non-availability of a full text (*n* = 5), manuscripts not in the English language (*n* = 9), case reports (*n* = 5), or expert’s opinion (*n* = 2). Articles concerning: (i) children with cranio-facial malformations (e.g., cleft lip palate); (ii) syndromic children (e.g., Marfan’s syndrome); (iii) data on dental characteristics and orthodontics techniques comparisons; and (iv) where RME was part of a combination of different orthodontic approach, i.e., RME + mandibular advancement devices (*n* = 392), were also excluded. 

A total of 155 papers were assessed in order to identify the papers suitable for the present review. Additional inclusion criteria for the clinical series were papers with an adequate group of patients studied (*n* > 20); for the review papers, contributions were considered when they were published in impact factor journals and showed rigorous methods and reporting. 

After the application of the additional evaluation criteria 112 papers (29 reviews and 83 case series) resulted as appropriate for this review. The selection process data are summarized in the flow chart in Figure 1. 

## 3. Results

The possible effects of the RME application in otolaryngology are summarized in the sections below. In each section, the most recent findings are reported. The selected population, within the analyzed papers, has an average age of 10.5 years (range 5–14).

### 3.1. RME as a Possible Alternative or Adjuvant Therapy to Adenotonsillectomy

Nine articles focused on RME in the management of adenoid/adenotonsillar hypertrophy. No reviews were available on this topic. 

The reported population in each of the nine articles was rather homogeneous (average age of 6.9 years), but the focus and the employed methods are often different. The main points addressed include: (i) an improvement in nasal breathing, if adenoidectomy/tonsillectomy is associated with RME; (ii) the best timing to introduce a RME (i.e., before adenotonsillectomy?); and (iii) RME as a possible alternative for patients requiring adenoidectomy. 

Most of the studies demonstrated an efficacy of RME on OSA and on nasal breathing, especially in patients with grade I–II tonsils of and/or small adenoids. 

Adenotonsillectomy in children is often the very first therapeutic approach against OSA; however, new data suggest that apneas could reappear later [2,3]. Two studies evaluated the combined treatment of adenotonsillectomy and RME against pediatric OSA. Villa M.P [4] and Guilleminault [5] suggested that an integrated therapy is helpful. In particular, Villa suggested that starting an orthodontic treatment as early as possible could be important in order to increase the treatment efficacy. Additionally, Di Vece et al. demonstrated that children with hypertrophic adenoids could benefit from RME, and thus could avoid an adenoidectomy operation [6].

Matsumoto et al. [7] assessing a group of children with hypertrophic adenoids, demonstrated that, in 44% of the cases where nasal resistance was evaluated by acoustic rhinometry and computerized rhinomanomety, a decrease in the rhinometry values was observed after the RME application without any adenoidectomy.

In addition, the combination of adenotonsillectomy and RME has been reported to be useful in the restoration of a balanced transversal, sagittal and vertical facial growth [2]. 

In any case, RME could eventually have positive effects on the reduction of adenotonsillar volume, while there is no evidence that RME could replace adenotonsillectomy. 

### 3.2. Effects of RME on Recurrent Otitis Media with Effusion and Conductive Hearing Loss in Children

Eight papers, including six case-series and two reviews, regarding the effects of RME on recurrent otitis media with effusion and conductive hearing loss (CHL) were considered.

Villano et al. [8] studied RME in 25 subjects with maxillary constriction and recurrent serous otitis media with conductive hearing loss (CHL). Only four patients showed an improvement in their audiometry and tympanometry data immediately after the expansion application. This improvement was also limited to the lower audiometric frequencies (250–1000 Hz). On the contrary, after using the expansion for 8 months, all tympanograms were of type A, and the hearing threshold returned to normal in all frequencies. This study demonstrates the importance of the RME application and the time requirements necessary for obtaining significant and stable results in the middle ear function.

Kilic et al. [9] compared the effects of RME and of a trans-tympanic tube placement on pure-tone threshold in children with recurrent otitis media with effusion (OME). They divided children into three groups: a control group, a ventilation tube placement group (children with resistant OME lasting at least 3 months and conductive hearing loss—CHL) and an RME group. The RME group consisted of children with resistant OME and CHL, in addition to a maxillary constriction, a deep palatal vault, and a bilateral crossbite. The improvement on hearing threshold after RME treatment was significantly greater than the one obtained using the ventilation tube. The data from this study suggested that, in patients with resistant OME and associated skeletal deficiency (maxillary constriction), RME could be an alternative to ventilation tubes. 

De Stefano et al. [10] studied the effects of RME in children affected by recurrent otitis media (ROM), adenoid hypertrophy and maxillary constriction. They reported that, after RME application, the nasal function, the auditory threshold and the tympanometry were all significantly improved. The benefits of RME application were also reflected in the subsequent reduction of the adenoid tissue, according to the Cassano’s classification: after 6 months of treatment, 20 patients with adenoid hypertrophy grade III and seven with grade II, became grade I (25 subjects) and grade II (two subjects), respectively. These results were stable at 12 months post treatment. Obviously, a criticism of this study is the subjective assessment of hypertrophic adenoids, by a clinical examination with nose-endoscopy. Only two patients out 27 (7.4%) required adenoidectomy after RME to restore Eustachian tube function [11]. 

Cozza et al. [12] studied RME in 24 patients with CHL due to otitis media and oral breathing with a pattern of an atypical swallowing. They showed that RME enlarged the nasal fossa and also reduced the nasal flow resistance. In 23 out of 24 patients, they reported a complete recovery from CHL and only one case presented a light CHL with a type C tympanogram. Main drawback of this study is the absence of a control group, in light that CHL could resolve itself within 6 months or could improve spontaneously by the child’s growth. 

Pirelli et al. [13] applied RME in 42 children affected by malocclusion and OSA, with a long follow-up (10–16 months). They showed (instrumentally by rhinomanometry, audiogram, tympanogram, and polysomnographic evaluations) a complete normalization of all CHL cases (eight patients), type-C tympanograms (11 patients), nose breathing deficits (34 cases) and OSA index (from 42 patients with AHI > 5), at the end of the study-period.

In 2014, Eichenberger and Baumgartner [14] reported the results of a review on the effects of RME on children with nasal breathing, OSAS, nocturnal enuresis and CHL, suggesting an improvement of conductive hearing thresholds. Their analysis was performed on the basis of five previous studies (Laptook et al. 1981, Ceylan et al. 1996, Taspinar et al. 2003, Villano et al. 2006, Kilic et al. 2008).

In 2017, Fernandes Fagundes [15] analyzed nine studies regarding the auditory improvement due to the RME application. Nine prospective nonrandomized clinical trials were selected. Eight of them reported an improvement in hearing levels, which were stable at the post-treatment follow ups.

Most of these studies (Cozza [12], Pirelli [13]) included children without evidence of an adenoidal blockage at the nasopharyngoscopic evaluation, eventually because of previous adenoidectomy or adenotonsillectomy (i.e., 73.8% of children included by Pirelli). Kilic [9] excluded children previously treated by adenotonsillectomy, but did not include information on adenoid dimension, such as Villano [8]. Conversely, De Stefano described, according to Cassano’s classification, the reduction in adenoidal tissue due to RME application. 

In any case, the effectiveness of RME on recurrent otitis media seems not to be related to a direct effect on the adenoidal volume; it is likely that RME application could possibly improve the Eustachian tube function by “stretching” the elevator and tensor palatine muscles and widening the median palatine suture.

### 3.3. Effects of RME on Nasal Breathing 

Much controversy still exists regarding the effects of RME in reducing nasal resistance, despite its proven effectiveness on nasal and maxillary expansion. 

According to Moss’s functional matrix concept, an adequate nasal breathing allows a proper growth and development of the craniofacial complex [16]. Thus, a continuous airflow through the nasal cavity during breathing represents a constant stimulus for the lateral growth of maxilla and for the lowering of the palatal vault. On the other hand, midface hypoplasia or constricted upper dental arch can lead to upper respiratory tract obstruction and oral breathing [17,18]. 

Many studies have evaluated the effects of RME application on nasal airway patency with objective tests such as rhinomanometry and acoustic rhinometry. 

Monini et al. [18] showed that, by using active anterior rhinomanometry measured both in supine and orthostatic position, 66% of patients post-RME application improved their nasal flow in the supine position, while 50% increased their nasal flow in the orthostatic position. This improvement was not limited to the first 10–14 days after RME application (early post-RME), but nasal resistances still improved in late post-RME (1 year post-RME), with an improvement of 65% in supine position and 91% in orthostatic position, respectively. 

The effects of RME on nasal resistance were evaluated with acoustic rhinomanometry [7,19,20,21,22,23] and in two meta-analyses [22,23]. 

Langer et al. [24], using computed rhinomanometry, showed a significantly decreased nasal resistance 30 months following RME application. 

Kabalan et al. [20] used both acoustic rhinomanometry (AR) and a cone beam CT (CBCT) to assess direct correlations between functional and dimensional changes in the airways before and after the appliance placement. AR measurements were obtained at baseline (T1) and 6 months after completed expansion at the time of the appliance removal (T2). Even though the minimum cross-sectional area (MCA) and nasal cavity volume (NCV) showed an improvement between T2 and T1, they did not find correlations between these results and the improvement in CBTC. 

An improved nasal function may enable the nasal filtrum action, avoiding mouth opening and oral breathing, and thus decreasing respiratory infections [25,26].

Ceroni Compadretti et al. found greater increases in MCA and nasal cavity volume (NCV) in the treated group compared to the control group, in basal conditions and after the use of a nasal decongestant [25].

On the other hand, Enoki et al. [19] did not detect significant differences in minimum cross-sectional area (MCA), either in the region of the nasal valve or in the inferior turbinate at the three time points studied. They conclude that the mucosal benefits of RME are not as evident as those on the maxilla-facial bony parts.

A similar conclusion was also reported by Matsumoto et al. [7]. However, despite the lack of significant differences in AR, the application of RME decreased nasal resistance during the first 3 months of evaluation, even if this result was not persistent.

Other studies added a fiberoptic evaluation to the AR; Enoki et al. [19] detected an inferior turbinate hypertrophy in 75.8% of the assessed patients (22/29) and a significant adenoidal obstruction (between 50 and 90%) in 37.9% (11/29) of them.

### 3.4. Effects of RME on Airway Dimensions Based on Cephalometric Measurements

Twenty case-series studies and three reviews focused on cephalometric measurements after RME application. The studies mainly differ in the radiological means they used. Six studies used both anteroposterior and lateral cranial scans. Three studies used only anteroposterior scans and eight only lateral scans. Moreover, three studies used cephalometric scans associated with orthopantomography (see also Table 1, Table 2 and Table 3).

These studies also differ in terms of the evaluation timing following the RME application. Two studies assessed the effects of this procedure immediately after its removal [19,26]. Enoki et al. [19] found a significant expansion of nasal cavity following the RME application, which remained stable after 3 months. No statistically significant differences were observed in acoustic rhinometry, only a lowering in nasal resistance values. Celikoglu et al. [26], using a lateral cephalography, suggested that significant increases in the upper pharyngeal dimension and in the vertical position of hyoid bone did occur.

Five studies evaluated the RME effects after 6–9 months [12,27,28,29,30]. Baldini et al. [27] compared two different expansion protocols and found an anterior movement of the mandible due to the RME application. Hoxha et al. [28] showed the presence of a significantly increased nasopharyngeal and pharyngeal area. Cozza et al. [2] found a significant increment in the dimensions of the nasal cavity, and no significant differences in the maxilla or mandible length. Manni et al. [28,31] found that the RME produced a significantly increased oropharyngeal, laryngopharyngeal, and nasopharyngeal space. Farronato et al. [29] found a significant backward and downward rotation of the palatal plane and a forward position of the mandible in all malocclusion classes. 

Ten studies evaluated the long-term effects of RME, from 12 to 36 months post application. Only one study evaluated patients in a 7–8 year follow-up period [3]. Six studies used anteroposterior cephalometric radiograph and reported a stable and statistically significant enlargement of the nasal cavity width, while four out of six studies found a statistically significant increase in intermaxillary width [25,32]. Pirelli et al. found an improvement in nasal flow by anterior rhinometry, while Ceroni Compadretti et al. found that the greater the diameters of the maxilla prior to RME were, the lower was the benefit on the nasal airway resistance [13,25]. 

Seven studies evaluated cephalometric measures on lateral cephalometric radiographs. Garib et al. [33] found that RME does not influence anteroposterior growth of maxilla and mandibula. Pereira et al., Monini et al., and Guilleminault et al. did not find an increase in cephalometric measures that express vertical facial growth due to RME [3,19,31]. Monini et al. and Eguren Langer et al. found an increased nasopharyngeal space [18,20]. Phoenix et al. and Ozbek et al. both found a reduction in the hyoid–mandible distance [32,34]. Ozbek et al. found a reduction in palate–tongue distances.

Lee et al. [35] focused on the changes in pharyngeal airway dimensions evaluated by lateral cephalographs and cone beam CT. In the lateral cephalographic radiographs, a significant increase was found in the anteroposterior after treatment. On the 3D CBCT, the nasal passage airway volume increased significantly more in the RME group than in the non- RME group. Barattieri et al. found an increased nasopharyngeal space and an increased nasal cavity width [22]. 

### 3.5. Effects of RME on Airway Dimensions and Volume Based on CBCT and 3D CT Evaluation

Computed tomography (CT) is the best way to compare the airway geometry before and after RME application; in particular, the Cone Beam Computer Tomography (CBCT) is considered the gold standard, especially in the pediatric population. CT analysis with software and 3D reconstructions allows quantification of width, area, and volume. Computational fluid dynamic on CT images allows the evaluation of the modification of nasal resistances and pharyngeal pressures.

Twenty-four studies evaluated the RME efficacy with CBCT. Five studies used 3D CT analysis [34,36,37,38,39]. These studies showed differences between the measured segments. These include vertical length of the maxilla, length of the nasal septum (on axial images) or width of nasal cavity. 

All the included studies showed that RME increased both lateral and vertical distances (see also Table 4).

### 3.6. RME and OSA 

The first study evaluating the effects of RME on OSA was published by Cistulli et al. in 1998. We found in the literature twenty-one studies that met the inclusion criteria and focused on the effects of maxillary arch expansion on OSA [43].

The American Academy of Pediatrics indicated polysomnography as the diagnostic method of choice for OSA in children; however, most of the studies evaluated the effectiveness of RME application on OSAS on the basis of questionnaires only [63]. 

Eight studies evaluated the effect of the RME treatment on the apnea hypopnea index (AHI), and three studies reported mean SpO2 values. These data reflect the difficulties in evaluating reliable studies on this topic. 

Generally, as seen in Figure 2, the application of RME has positive effects on the obstructive sleep apnea in children and on the AHI index. Villa et al. showed that RME is an effective treatment for children with OSA [64]. This study suggested that the positive effects of RME on OSA resulted from increased pharyngeal dimensions, a new tongue posture, a changing of anatomical structures, an improved nasal airflow, a significant progress of naso-pharyngeal functions, and reduced naso-respiratory problems. Cistulli et al. speculated that the mechanism for OSA improvement, after RME application, is related to the improved nasal airflow, which results in the generation of lower sub-atmospheric inspiratory pressures and hence reduces the possibility to collapse of pharyngeal walls [58].

### 3.7. RME and the Voice

The expansion of the airway size—in terms of palatal and nasal width, total volume, and modification of its shape—induced by the RME can modify the quality and resonance of the voice. The RME can also affect the tongue posture and the palatal volume. In this context, lots of studies have performed voice recordings before and after RME application.

Five studies focused on the changes in the voice due to RME [66,67,68,69,70]. Biondi and colleagues showed that there was an impact on the voice not only after RME, but also during the application. RME can cause modification of both fricatives and the vowel sound /i/, while palatal consonants usually do not change significantly [71]. The modifications correspond to a reduction in the volume of resonance cavities after RME, confirming that the tongue moves higher in the oral cavity, closer to the palate. 

Bilgic F et al. [67] showed that, after RME, the jitter percentage was significantly decreased, while the shimmer percentage was significantly increased, concluding that RME could significantly change the voice quality. 

Macari et al. [69] showed no differences post RME on the average fundamental frequency (F0), but a lowering of the means of formant frequencies F1 and F2. The latter were recorded by asking each individual to pronounce and sustain, at a comfortable pitch and intensity level, the vowel sounds /α/.

Yurttadur et al. [68] showed that the RME group presented a decrease in both F1 and F2 frequencies and an increase in the F3 frequency, but these differences were not statistically significant. 

## 4. Discussion and Conclusions

Otolaryngologists should be aware of all of the indications, benefits and potential implications of the RME treatment. As reported in the previous sections, the RME represents not just a precise orthodontic instrument, but also an integrated treatment able to improve nasal function, potentially sparing ventilation tube placement and eventually improving adeno-tonsillar hypertrophy, therefore also having positive implications for conductive hearing loss and OSA. 

The improvement of nasal breathing seems to be related to different factors, and particularly to the widening of maxillofacial spaces and of airway size. In this sense, recently, Yoon et al., 2022 [70] reported that RME in children could reduce both adenoidal and tonsillar volume. In particular, as a result of RME application, they described that 90% of children had a reduction of adenoidal volume, and that 97.5% had a reduction of tonsillar volume, with average decreases, measured by CBTC, of 20.1% and 40.2%, respectively. Furthermore, other described positive effects of RME application on the nasal function have been related to the enhanced nasal filtrum action and to the reduction of oral breathing, with a consequent decreased incidence of recurrent naso-respiratory infections [25,26]. 

From this perspective, as evidenced by many authors, the application of RME can have positive effects on OSA. Several mechanisms have been advocated, such as (i) the reduction of adeno-tonsillar volume, (ii) the reduced incidence of tongue collapse due to persistent oral breathing, and (iii) the stiffening of the collapsible pharyngeal segment of the airway. 

Moreover, the application of RME could improve the Eustachian tube function, favoring the function of the elevator and tensor palatine muscles with possible positive effects on recurrent otitis media with effusion and conductive hearing loss in children.

Hence, otorhinolaryngologists should routinely carefully consider the maxillary conformation of their little patients, since the individuation of an altered skeletal structure can lead to the right treatment, avoiding over-treatment choices (i.e., adenotonsillectomy or tympanic ventilation tube placement). In fact, a condition characterized by maxillary constriction, posterior crossbite, high palatal vault, elevation of nasal floor and mouth breathing (such as the skeletal development syndrome, first described by Laptook in 1981), could produce different detrimental effects on the whole maxillofacial development [71]. 

In our opinion, the beneficial restoration of nasal breathing after the application of RME should be evaluated by larger studies. It is likely that further studies could demonstrate that early application of RME at pediatric age could also avoid septal deviation and septal surgery at adult age. 

The application of RME has been reported to be more effective in younger patients, before the maxillary sutures are fully ossified, until the age of 14–15 in females and 15–16 in males [72]. However, since RME application requires patient compliance and the eruption of an adequate number of teeth to support the device, children undergoing orthodontic treatment should be carefully evaluated. 

Furthermore, the effects of RME seem not be limited to the head and neck regions. Several studies have investigated different issues such as nocturnal enuresis (NE), reporting that the RME application can improve resistant enuresis [14,49,60,73,74,75]. The explanation given by Bazargani and colleagues is that the positive influence of RME application on enuresis could be linked to the fragmentation of sleep architecture due to the OSA [75]. 

The present study is not free from various drawbacks and limitations, such as (i) the difference in the number of patients enrolled in each analyzed paper; and (ii) the presence of different methods/tools in the clinical assessment of the enrolled patients. 

In conclusion, otolaryngologists should be aware of all the indications, benefits and potential implications of the RME treatment. The application of RME should be considered carefully, particularly evaluating the possible multiple effects not just on maxillofacial development, but also on middle ear function, OSA, enuresis, and the voice. A multidisciplinary approach is always recommended for the diagnosis and treatment of these problems.

## Figures and Tables

**Figure 1 jcm-11-05243-f001:**
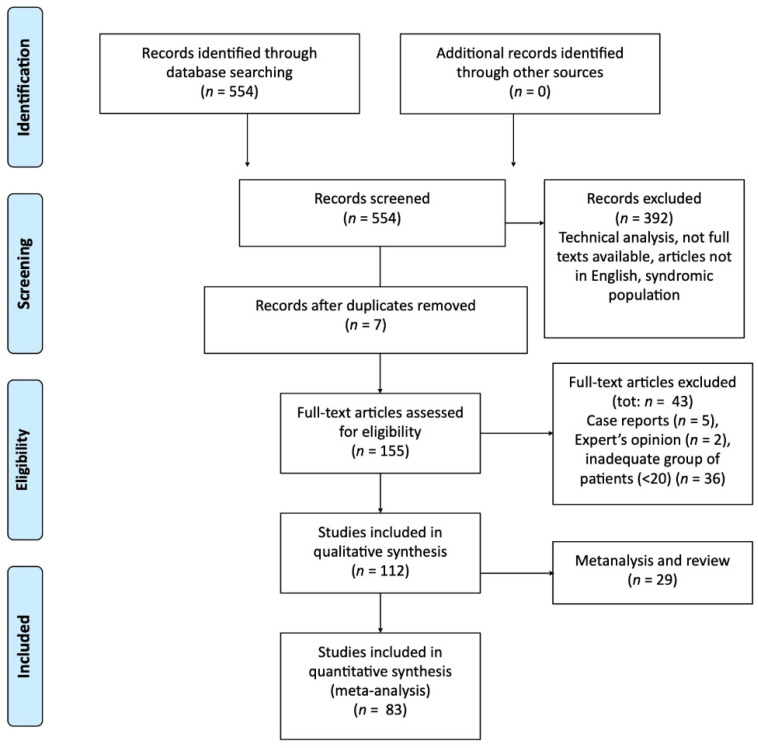
Literature evaluation and selection, according to PRISMA criteria (http://www.prisma-statement.org/ (accessed on 16 July 2022).

**Figure 2 jcm-11-05243-f002:**
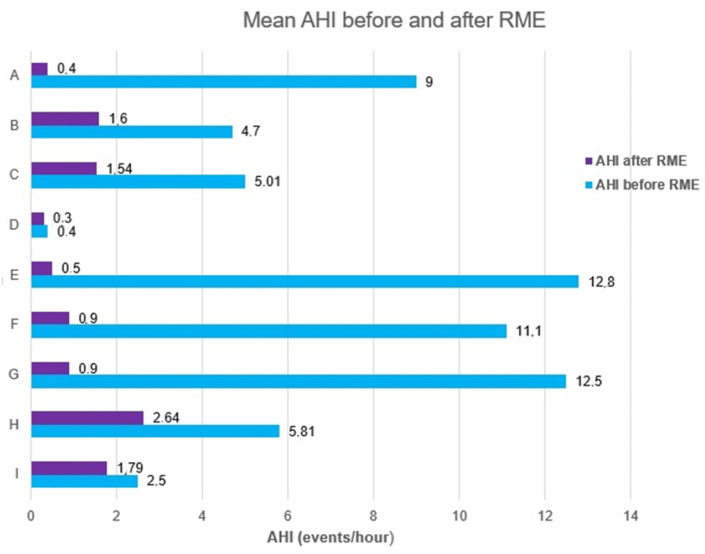
Reported effects on the Apnea Hypopnea Index (AHI) index after Rapid Maxillary Expansion (RME) application. A: Guilleminault C. et al., 2013 [3]. B: Villa M.P. et al., 2015 [64]. C: Fastuca R. et al. [56]. D: Pirelli P. et al., 2015 [65]. E: Pirelli P. et al., 2005 [13]. F, G: Guilleminault C. et al., 2011 [5]. H: Villa M.P. et al., 2013 [4]. I: Hoxa S et al. [28].

**Table 1 jcm-11-05243-t001:** Effects of RME on airway dimensions and volume: anteroposterior scans vs. lateral width.

*Anteroposterior Scans—Lateral Width*	
Maxilla width	cross-sectional linear measure between the maxillary tuberosity and the contour of the zygomatic apophysis)
Nasal width	cross-sectional linear measure of nasal cavity from right lateral wall to left lateral wall
UNR-UNL: (UN) upper nasal points, right (R) and left (L)	inner points on the nasal opening taken parallel to the HRP (horizontal reference plane)
LNR-LNL, right and left lower nasal (LN) points	lateral points on the nasal opening taken parallel to the HRP
MXR-MXL, right and left maxillary (MX) points	deepest points on the curvature of the maxillary malar process
U6R-U6L, right and left upper first molar (U6) points	midpoint on the buccal surface of the maxillary first molar crown

**Table 2 jcm-11-05243-t002:** Effects of RME on airway dimensions and volume: lateral scans vs. vertical growth.

*Lateral Scans—Vertical Growth*	
SN.Gn	angle between the Saddle, Nasion and Gnatio points
SN.GoMe	angle between the anterior skull base (SN) and the mandibular plane (GoMe)
ML-NL	angle between mandibular line and nasal floor line
SNA, maxillary angle	angle by the lines Sella-nasion and nasion-A (A indicates the most retracing part of the maxilla)
SNB, mandible angle	angle between the Sella-nasion and nasion-B lines (B indicates the most retracting part of the mandible)
ANB, skeletal class	angle between the lines A-nasion and nasion-B
FMA	angle between the Frankfurt plane and the mandibular plane
SPG (superior pharyngeal gradient)	PNS (posterior nasal spine)-So (Sella midpoint)/PNS-d2 (intersection on posterior pharyngeal wall)
ING (inferior nasopharyngeal gradient)	PNS-Ba (Sella basal point)/PNS-d1 (intersection on posterior pharyngeal wall)
H-MP	perpendicular distance from hyoid to mandibular plane
H-ANS-PNS	perpendicular distance from hyoid to palatal plane
Tongue-Palatal distances	distance between the hyoid bone and the upper tongue point
Co-Gn	linear distance between the condyle point and the Gnatio point
A/Olp, Maxillary bone base	distance between the Olp line (line perpendicular to occlusal line) and point A
PG/Olp, mandibular bone base	distance between the Olp line (line perpendicular to occlusal line) and point PG (pogonion)
MP^PP	angle between mandibular and palatal plane
SN^PP	angle between cranial base and palatal plane
Inter inc	interincisive angle
IsiP^SN	angle between cranial base and upper central incisor
IiiP^MP	angle between mandibular plane and lower central incisor
N-Me (anterior face height)	distance between Nasion and the lowest mandible point
ANS-Me (lower anterior face height)	distance between anterior nasal spine and lowest mandible point
S-Go (posterior face height)	distance between sella and mandible angle point
OVJ, overjet	vertical distance between incisive margin on upper and lower incisives
OVB, overbite	horizontal distance between incisive margin and lower incisives

**Table 3 jcm-11-05243-t003:** Effects of RME on airway dimensions and volume: lateral scans vs. pharyngeal measures.

*Lateral Scans—Pharyngeal Measures*	
Antero-posterior nasopharyngeal area (ad1-ad2-PNS)	area between the PNS—posterior nasal spine-, ad1—intersection of the PNS-So line to the posterior wall of the nasopharynx- and ad2—intersection of the PNS-Basio line to the posterior wall of the nasopharynx
PNS-AD1	distance between the posterior nasal spina (PNS) and the posterior pharyngeal wall along the line from PNS to basion (Ba)
AD1-Ba	distance between basion and adenoid (AD1) along the line from PNS to basion
PNS-AD2	distance between PNS and adenoid tissue (AD2) along the line from PNS to Hormion (H, the point located at the intersection between the perpendicular line to Sella-Ba and the cranial base)
AD2-H	distance between AD2 and H
PNS-Ba	distance between PNS and Ba
Ptm-Ba	distance between pterygomaxillare (Ptm) and Ba
PNS-H	distance between PNS and H
McNamara’s upper pharyngeal dimension	distance between the soft palate and the nearest point of the posterior pharyngeal wall
McNamara’s lower pharyngeal dimension	distance between the posterior tongue contour and the pharyngeal wall
Lower pharyngeal dimension	distance between the anterior and posterior pharyngeal wall through the line between the anteroinferior edge of C3 and the posteroinferior edge of C3
Laryngopharyngeal airway space LA	Linear distance of the laryngopharyngeal space along the C4 plane

**Table 4 jcm-11-05243-t004:** Reported effects of RME on airway dimensions and volume.

		*Studies Showing a Dimensional Increase*	*Studies Showing a Dimensional Decrease*	*No Differences*
**Volume**	**Nasal cavity**	(independently of RME type), (not influenced by age, sex, skeletal class) [38,39,40,41,42,43]		
	**Nasopharynx**	[39,40,44]		
	**Oropharynx**	[3,42,44]	(lowering anterior palate), [39,45]	[40]
	**Maxillary sinus**	(RME, not SME) [36,42,46]		
	**Orbital**	[47]		
	**Intraoral**		[44]	
	**Total**	(the lower the age, the better the result), [42,44,48]		[49]
**Distances**	**Vertical of the maxilla**	[37,39,48,50]		
	**Antero-posterior (nasal septum length)**	[37,50,51]		
	**Maxillary width**	(independently of RME type) [38,40,41,52,53]		
	**Intermolar width**	(independently of RME type) [36,51,54,55,56]		
	**Nasal width**	[41,52,57]		[41]
	**Palatal width**	[41,45,58,59,60]		
	**Soft tissues**	(both prepuberal and postpuberal), [61,62]		
**Areas**	**Mandibular**	[54]		

Da Luz Baratieri et al. [22] showed improvements on the airway dimensions and the nasal breathing in patients treated by RME. Luebbert J. et al. showed that dental changes are more significant than skeletal changes [53].

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
