# Peer review of "Rapid Maxillary Expansion (RME): An Otolaryngologic Perspective"

_jcm, 2022, doi:10.3390/jcm11175243_

Round 1

Reviewer 1 Report

This study reviews the association between RME and ENT, which helped ENT physicians understand how this orthodontic procedure was involved in the management of ENT disease. This was the novelty of this study. However, based on current research, there was insufficient evidence to recommend RME as an alternative to classic ENT therapy.

There were some problems in the manuscript.  

1) In figure 1. “inadeguate group” should be “inadequate group”? “Metanalisis” should be “meta-analysis”?

2) In the section of results, some summaries or descriptions about the results were controversial. Page 3 line 78. “3.1 RME as a possible alternative or adjuvant therapy to adenotonsillectomy”. The references cited in the manuscript did not provide sufficient evidence that RME could replace adenotonsillectomy. It had only been demonstrated that RME could be applied alone or in combination with adenotonsillectomy to improve the nasal breathing in pediatric patients with OSA. The comparison of treatment effect between RME and adenotonsillectomy did not be mentioned in the study. 

3) In the section of "3.2 Effects of RME on recurrent otitis media with effusion and conductive hearing loss in children". The results did not describe whether children receiving RME had hypertrophic adenoids. It is well known that adenoid compression is the main cause of CHL in children. Therefore, adenoidectomy is usually considered first rather than RME, and has achieved satisfactory effect in most pediatric patients. It should explain why RME was selected for CHL in these studies, and what was the adenoid grade and history of adenoidectomy in these patients. That will help the readers understand potential advantages of RME in the treatment of CHL.

4) In the section of 3.3. The aim of this part was to demonstrate the effect of RME on nasal resistance. The title "Relationships among maxillary constriction, mouth breathing and nose-respiratory problems" was too broad. The references in this section did not discuss these complex relationships.    

5) The section of 3.4 and 3.5 was similar. Hence, the title of 3.4 should be “Effects of RME on airway dimensions based on cephalometric measures”. The title of 3.5 should be “Effects of RME on airway dimensions and volume based on CBCT and 3D CT evaluation”.

6) Page 10 line 314, the RME might be an integrated treatment for some ENT diseases, but it was not yet a substitute for other traditional ENT surgeries.

7) The section of discussion is oversimplified. As the author mentioned in the first line of this paragraph, more discussion should be added to show the readers that which disease is suitable for RME; what are the advantages and disadvantages of RME compared to traditional ENT treatment; why RME has better results than other ENT procedures in some diseases, such as pediatric OSA, recurrent otitis media, and what the mechanisms are.

Author Response

1) In figure 1. “inadeguate group” should be “inadequate group”? “Metanalisis” should be “meta-analysis”?

Thank you for your comments: figure 1 has been modified accordingly.

2) In the section of results, some summaries or descriptions about the results were controversial. Page 3 line 78. “3.1 RME as a possible alternative or adjuvant therapy to adenotonsillectomy”. The references cited in the manuscript did not provide sufficient evidence that RME could replace adenotonsillectomy. It had only been demonstrated that RME could be applied alone or in combination with adenotonsillectomy to improve the nasal breathing in pediatric patients with OSA. The comparison of treatment effect between RME and adenotonsillectomy did not be mentioned in the study.

Thank you for this comment; we have specified this point in the text, as suggested. In particular we have stated that RME could eventually have positive effects on the reduction of adenotonsillar volume, while there is no evidence that RME could replace adenotonsillectomy.

3) In the section of "3.2 Effects of RME on recurrent otitis media with effusion and conductive hearing loss in children". The results did not describe whether children receiving RME had hypertrophic adenoids. It is well known that adenoid compression is the main cause of CHL in children. Therefore, adenoidectomy is usually considered first rather than RME, and has achieved satisfactory effect in most pediatric patients. It should explain why RME was selected for CHL in these studies, and what was the adenoid grade and history of adenoidectomy in these patients. That will help the readers understand potential advantages of RME in the treatment of CHL.

Thank you for this comment; we have now specified this point in the text, as suggested, by including a new paragraph.

4) In the section of 3.3. The aim of this part was to demonstrate the effect of RME on nasal resistance. The title "Relationships among maxillary constriction, mouth breathing and nose-respiratory problems" was too broad. The references in this section did not discuss these complex relationships.   

Thank you for this comment; we have now modified the title as requested: ‘Effects of RME on nasal breathing’.

5) The section of 3.4 and 3.5 was similar. Hence, the title of 3.4 should be “Effects of RME on airway dimensions based on cephalometric measures”. The title of 3.5 should be “Effects of RME on airway dimensions and volume based on CBCT and 3D CT evaluation”.

Thank you for your comment; we have now modified the titles 3.4 and 3.5 as suggested.

6) Page 10 line 314, the RME might be an integrated treatment for some ENT diseases, but it was not yet a substitute for other traditional ENT surgeries.

Thank you for the comment; we have now specified that RME might be an integrated treatment for some ENT diseases, in selected cases.

7) The section of discussion is oversimplified. As the author mentioned in the first line of this paragraph, more discussion should be added to show the readers that which disease is suitable for RME; what are the advantages and disadvantages of RME compared to traditional ENT treatment; why RME has better results than other ENT procedures in some diseases, such as pediatric OSA, recurrent otitis media, and what the mechanisms are.

Thank you for this comment. We have now added a new paragraph to the discussion section in order to address these issues.

Reviewer 2 Report

The paper is unclear as to whether or not this review applies to the pediatric or adult population (or both?) the subject patient population is not well defined. 

Reasonable review of the pediatric literature related to RME and OME

Table 4 can be organized in a more straightforward manner. What is the difference between "studies that didn't show any improvement" and "No Differences?" 

Review of RME as it relates to sleep disordered breathing is cursory and incomplete as this is the major non-occlusal indication for maxillary expansion. Recommend review of literature published by Liu SY, Yoon A et al as it relates to Maxillary Expansion in SDB

Frequent spelling / Grammatical errors related to english translation make the text difficult to follow in places 

Author Response

The paper is unclear as to whether or not this review applies to the pediatric or adult population (or both?) the subject patient population is not well defined.

Reasonable review of the pediatric literature related to RME and OME

Thanks for your comment. We have now cleared this point either in the introduction, either within the results section, in particular stating that the selected population, within the analysed papers, has an average age of 10.5 years (range 5-14).

Table 4 can be organized in a more straightforward manner. What is the difference between "studies that didn't show any improvement" and "No Differences?"

Thanks, we have now cleared this point in table 4: in particular, we have now modified the column headings into: studies showing a dimensional increase (of volume, distances or areas), studies showing a dimensional decrease, and studies showing no differences.

Review of RME as it relates to sleep disordered breathing is cursory and incomplete as this is the major non-occlusal indication for maxillary expansion. Recommend review of literature published by Liu SY, Yoon A et al as it relates to Maxillary Expansion in SDB

Many thanks: section 3.6 of the results is all related to RME and OSA in children. Moreover, most of the papers by Liu SY, Yoon A refers to (i) the maxillary expansors in adults and (ii) to the application of DOME (distraction osteogenesis maxillary expansion) and not to RME. In any case, we have now cited within the discussion section the latest paper by Yoon A et al (2022) about the relation, in children, between RME and adenotonsillar hypertrophy.

Frequent spelling / Grammatical errors related to english translation make the text difficult to follow in places

The paper has now been reviewed by the native English language speaker at our Department.

Reviewer 3 Report

This study is a literature review to investigate the effects of RME on otolaryngologic cases. The experimental strategies are well designed and overall, the data are clear presented. However, the abstract is poorly written, and it should be add more information and the details to show the study clearly.

Author Response

This study is a literature review to investigate the effects of RME on otolaryngologic cases. The experimental strategies are well designed and overall, the data are clear presented. However, the abstract is poorly written, and it should be add more information and the details to show the study clearly.

Thank for your comment. We have now modified the abstract accordingly, in order to make it more informative.

Round 2

Reviewer 1 Report

In the results, the author mentioned that the selected population within the analyzed papers has an average age of 10.5 years (range 5-14). Dose that means all children over the age of 5 can undergo RME? I suggest adding a section on RME timing-selection to the discussion. Since RME has positive effects on some ENT diseases, otolaryngologists should learn from this manuscript about the appropriate timing of RME for different conditions, such as pediatric OSA, CHL, mouth breathing.

Author Response

In the results, the author mentioned that the selected population within the analyzed papers has an average age of 10.5 years (range 5-14). Dose that means all children over the age of 5 can undergo RME? I suggest adding a section on RME timing-selection to the discussion. Since RME has positive effects on some ENT diseases, otolaryngologists should learn from this manuscript about the appropriate timing of RME for different conditions, such as pediatric OSA, CHL, mouth breathing.

Thank you for this comment. As requested, we have further specified in the discussion section that, as reported in the literature, the application of RME has been indicated to be more effective in younger patients, until maxillary sutures are not fully ossified, until the age of 14–15 in females and 15–16 in males.

English language: spell check required.

Thanks: the paper has now been further reviewed by a native English language speaker.

Reviewer 2 Report

Appropriate and balanced revisions 

Author Response

English language and style are fine/minor spell check required

Thanks: the paper has now been further reviewed by a native English language speaker.